# Body Composition Trend in Slovene Adults: A Two-Year Follow-Up

**DOI:** 10.3390/nu16234123

**Published:** 2024-11-28

**Authors:** Boštjan Jakše, Zlatko Fras, Uroš Godnov

**Affiliations:** 1Independent Researcher, 4280 Kranjska Gora, Slovenia; 2Centre for Preventive Cardiology, Division of Medicine, University Medical Centre Ljubljana, 1000 Ljubljana, Slovenia; zlatko.fras@kclj.si; 3Faculty of Medicine, University of Ljubljana, 1000 Ljubljana, Slovenia; 4Department of Computer Science, Faculty of Mathematics, Natural Sciences and Information Technologies, University of Primorska, 6000 Koper, Slovenia; uros.godnov@gmail.com

**Keywords:** adults, population-based, body mass index, body composition, body fat, fat-free mass, obesity classification, follow-up

## Abstract

This research re-evaluates the prevalence of obesity in a cohort of 432 Slovenian adults two years after an initial population-based cross-sectional examination, utilizing the World Health Organization’s body mass index (BMI) and total body fat percentage (FAT%) classifications. Herein, a medically approved electrical bioimpedance monitor was used to analyse body composition, and the results revealed a significant increase in the percentage of individuals classified as having overweight or obesity on the basis of BMI, from 40.7% initially to 45.2% at follow-up (FU); however, these percentages were notably higher in males than in females. The increases in body mass, BMI, fat mass, and FAT% were observed mainly in adult females, whereas in adult males, the increases in BM and BMI were attributed to fat-free mass (FFM) rather than fat mass. In this study, BMI was also compared with the FAT% obesity classification, and the BMI classification was shown to result in significantly fewer participants with obesity in both measures. However, the proportion of participants with obesity based on the two classifications did not differ substantially between the initial and FU measurements. In this study, mixed linear models were used to analyse overall trends and subgroup variations and highlight the importance of considering sex, age, and time of measurement when assessing body composition metrics. These findings emphasize the need for personalized health assessments and the importance of measuring body composition to evaluate adult obesity more accurately for both clinical assessments and public health policies. The state, in collaboration with social initiatives and industrial stakeholders, should prioritize these data and advance public health through innovative educational and awareness initiatives that are founded on robust scientific principles and that empower and promote the adoption of a healthy, active lifestyle.

## 1. Introduction

Obesity is a major public health challenge, and its global incidence has been increasing for decades [1,2]. This increase has been driven by urbanization, sedentary lifestyles, inadequate sleep, and high-calorie processed foods [3]. Overweight and obesity are associated with numerous chronic diseases. In the next 30 years, individuals with overweight are expected to account for 60% of patients newly diagnosed with diabetes, 18% of patients with cardiovascular disease, 11% of patients with dementia, and 8% of patients with cancer [4,5]. Among individuals aged 30–70 years, overweight is predicted to cause approximately 3 million premature deaths annually [6]. Obesity also leads to significant healthcare costs and a loss of productivity [7,8]. Compared with individuals who are overweight, individuals with obesity have 32% higher healthcare expenses and 68% higher indirect costs [7]. Preventable chronic diseases related to obesity are predicted to impose a global financial burden of 44 trillion euros as of 2023 [9].

According to the World Health Organization (WHO), approximately 59% of adults in Europe are classified as having overweight or obesity, highlighting a concerning trend [10]. If current trends continue, it is projected that by 2025, 33 of 52 Organization for Economic Cooperation and Development (OECD) countries will have an obesity prevalence of 20% or more [6]. Importantly, Slovenia has the highest prevalence of adult obesity (20.8%) among 20 European countries [10]. The Slovenian National Dietary Survey (SI.Menu) was conducted from 2017 to 2018 and included a sample size of 780; the results revealed that 59% of adults and 74% of elderly individuals had overweight or obesity [5]. Additionally, the initial study revealed that 42% of adults and 64% of elderly individuals had overweight or obesity [11]. Accordingly, between 1989 and the late 2000s, the percentage of Slovene children classified as having obesity tripled for both sexes, whereas the percentage of children with overweight doubled. Additionally, between 1989 and 2019, subcutaneous fat (rather than muscle mass) increased in children of all ages and sexes [12,13]. These high prevalence rates of overweight and obesity likely occur because fewer than 5% of adult Slovenes are healthy [14], and only approximately 10% of adult and adolescent Slovenes maintain a recommended diet with respect to the intake of fibre, vitamin D, and folate [15,16,17], as well as various plant-based food groups, such as vegetables, whole grains, potatoes, legumes, nuts, and seeds [18]. These statistics underscore the urgent need for effective interventions against obesity [19,20].

Body mass index (BMI) categorizes individuals on the basis of body height (BH) and body mass (BM), but BMI is not a reliable measure of total body fat percentage (FAT%). In addition, BMI does not distinguish between lean body mass (LBM) and fat mass (FATM), thus leading to possible inaccuracies [11,21,22,23]. Since BMI does not account for the FATM distribution [23], health authorities advise against its use alone for assessing overweight or obesity [24,25]. The use of BMI as a measure for estimating the prevalence of obesity may also be inaccurate, thereby hindering prevention and control efforts [26]. BMI should not be the only factor considered when assessing eligibility for bariatric surgery or semaglutide use [27]. Low muscle mass may present a greater health risk than high FATM [28]. Research in the U.S. has shown that BMI is not a dependable predictor of cardiometabolic disease and that it leads to the misclassification of approximately 30% of adults [29]. Thus, focusing only on BMI and BM reduction might not provide substantial health benefits and could contribute to weight-related discrimination and other harms [28].

Given the limitations of BMI, integrating BMI with body composition measurements is valuable, especially for studies that are smaller and/or focus on populations with a higher prevalence of sarcopenic obesity [21,30]. In our initial analysis of data used in this study, with a sample size of 844, proportions of obesity were also compared via BMI and FAT%, revealing that BMI underestimated the prevalence of obesity in females (13%) compared with the use of FAT% (17%) [11]. The secondary analysis explored leg FAT% and its relationship with FAT% and trunk FAT% according to the WHO classification’s BMI obesity cut-off values (Table 1), sex, and age. Additionally, linear regression revealed that variations in FAT%, sex, and age explained 82.5% of the variation in leg FAT% [31].

More than thirty years ago, the U.S. National Institutes of Health classified males with a BMI of 27.8 kg/m^2^ and females with a BMI of 27.3 kg/m^2^ as overweight. Individuals with BMIs below these values were considered ‘normal’ on the basis of an 85% cut-off from the National Health and Nutrition Examination Study II. In 1997, the International Obesity Task Force introduced ‘pre-obesity’ and a detailed BMI system: class I obesity (30–34.9 kg/m^2^), class II obesity (35–39.9 kg/m^2^), and class III obesity (≥40 kg/m^2^) [23]. In 1998, the U.S. Obesity Task Force set the normal–overweight BMI threshold to 25 kg/m^2^ and eliminated obesity subdivisions [34]. Importantly, different ethnic groups, such as Asians, may have lower BMI cut-off values. Furthermore, various guidelines caution against the universal application of BMI as a metric [27]. Therefore, some scholars have proposed focusing on fat-free mass (FFM) and body composition, asserting that the sole use of BMI and FAT% is insufficient to assess obesity-related health risks [28].

The primary objective of this two-year follow-up (FU) study was to assess trends in BM/BMI, body composition, and obesity prevalence on the basis of the WHO’s BMI and FAT% classifications among participants from the initial study [11]. The hypothesis that rates of overweight and obesity increase, particularly among adult females, was tested. As a secondary objective, obesity prevalence was compared between the initial and FU studies via BMI and FAT% cut-off values, and linear mixed models were employed to account for trends and subgroup variations.

## 2. Materials and Methods

### 2.1. Study Design and Eligibility

This cross-sectional study was approved by the Slovenian Ethical Committee in the Field of Sports on 26 June 2022 (approval no. 033–41/2022–5), and it was registered on 30 June 2022 at https://clinicaltrials.gov (document no. NCT05438966) [35]. To initially recruit participants, representatives from hotels, local governmental agencies, and fire stations that facilitate events on a national scale (such as conferences and meetings) were contacted. Additionally, social media platforms were utilized to ascertain the locations, dates, and types of complementary public programs (including seminars, workshops, training sessions, lectures, and courses) that did not require registration fees. In exchange for the collection and analysis of anonymized data from these measurements, organizers were provided with a complimentary body composition analysis, including interpretation and electronic printouts distributed to participants via email. Initial measurements occurred from 1 July to 31 August 2022 [11,35], with FU measurements taken in June 2024. Feedback on body composition was emailed as PDF printouts during the initial study, and completed questionnaires pertaining to dietary habits were collected [11]. The participants received email invitations for the FU and completed a brief demographic and lifestyle questionnaire via email. The FU initially included randomly recruited participants. No monetary incentives were offered in exchange for participation. The study was conducted in Slovenia, which has a population of 2.1 million. All participants were thoroughly briefed and asked to provide renewed consent.

### 2.2. Subjects

Healthy adults were invited to participate. There were no restrictions regarding the BMI of the participants. However, athletes, pregnant or lactating females, individuals with active chronic diseases, and individuals taking medication for chronic conditions (such as hypertension, hyperlipidaemia, diabetes, and thyroid disorders) were excluded. Those who recently changed their physical activity and diet were not excluded from the FU, as was the case during the initial study, aiming to minimize baseline measurement variability. Two pregnant females, two lactating females, and seven males who recently began taking hypertension and hyperlipidaemia medication were excluded from the FU study, which was conducted in six Slovenian cities/towns: Kranjska Gora, Kranj, Ljubljana, Novo Mesto, Maribor, and Šentilj. Therefore, the FU study included 432 adult Slovenes, accounting for 51% of the original participants (*n* = 844) [11].

### 2.3. Outcomes

#### 2.3.1. Body Mass Index and Body Composition Status

This study aimed to compare adult Slovene participants’ BM/BMI and body composition at two time points and aimed primarily to identify any significant demographic changes [11]. A calibrated 8-electrode bioelectrical impedance analysis (BIA) monitor (Tanita 780 S MA, Tokyo, Japan) [36] was used for all the measurements. Body height (BH) was measured with a floor scale (Kern, MPE 250K100HM, Kern & Sohn, Balingen, Germany) only during the initial study. Body mass index was calculated by dividing the BM by the square of the BH in metres. The body composition monitoring data included BM, BMI, FAT%, FATM, FATM per unit area of BH (FATM/BH) in kg/cm^2^, FFM in percentage and kg, FFM per unit area of BH (FFM/BH) in kg/cm^2^, total body water (TBW), and phase angle (PhA).

Body composition indices were compared with the WHO’s BMI and FAT% obesity targets. Obesity rates were examined on the basis of BMI (≥30 kg/m^2^) and FAT% (>35% for females, >25% for males) criteria [32,33]. The data were compared with the initial study results [11].

The findings were also delineated on the basis of age categories. The definition of ‘adult’ differs across sources and is frequently subject to inconsistent discourse [37,38,39]. In this analysis, the categorization of age was undertaken by delineating three distinct cohorts: (1) young adults, comprising individuals aged 18–39 years; (2) middle-aged adults, consisting of those aged 40–64 years; and (3) older adults, including persons aged 65 years and above.

In addition, linear mixed models were used to capture overall trends and subgroup variations, which are useful for data collected in hierarchical or nested structures, such as repeated measurements from the same participants. Due to the limited sample size of older adults, the focus of the study was predominantly on reporting findings for an ‘adult’ cohort (encompassing the young and middle-aged adult cohorts in our analysis). Nevertheless, the Appendix A present comprehensive data encompassing the entire sample, including data from older adults. Additionally, analyses of BMI and body composition changes were stratified into age and sex categories on the basis of the ‘adult’ and ‘older adult’ cohorts and age and sex categories, incorporating further distinctions on the basis of age classifications relevant to the discourse on ‘normal BMI’.

#### 2.3.2. Participant Characteristics

A concise sociodemographic and lifestyle questionnaire devised by the authors was used to gather essential participant data, including date of birth, residential region, education level (ranging from elementary to postgraduate), smoking status (categorized as current, never, or former), and frequency of physical activity (classified as fewer than three times per week or three or more times per week). Due to constraints in terms of human resources and time, it was not feasible to associate the basic demographic questionnaire directly with individual participants. Consequently, the data were automatically exported from the BIA via the manufacturer’s software. This limitation precludes the determination of whether increased physical activity correlates with improvements in body composition, thereby obstructing the establishment of meaningful correlations and comprehension of the impact of physical activity on health outcomes.

### 2.4. Statistical Analysis

Data preparation: All the statistical analyses were conducted via R version 4.3.1 and RStudio version 2024.04.1 Build 748. RStudio, an integrated development environment for R, provides a comprehensive suite of tools and packages for robust and efficient data handling and analysis, thereby enhancing the analysis. The tidyverse package [40] was utilized for data manipulation and visualization, as it has a cohesive framework for an efficient workflow. The janitor package [41] facilitates data cleaning, generating tidy datasets that are ready for analysis.

Descriptive statistics: The Arsenal package was used for detailed key study variable summaries and statistics [42]. The data were comprehensively described with means, medians, standard deviations, and CIs for continuous variables and with percentages and CIs for categorical variables. Additionally, the ggstatsplot package [43] was used to perform statistical analyses, such as the Mann–Whitney U and Kruskal–Wallis tests.

Analysis of differences in obesity: To assess differences in obesity proportions between BMI and FAT%, a two-sample test with continuity correction was used for large sample sizes. The *p* values were adjusted with the Benjamini-Hochberg method to control the false discovery rate and reduce type I errors.

Parametric and nonparametric tests: Continuous data normality was assessed via the Shapiro-Wilk test. If the data followed a normal distribution, paired *t* tests were used to compare means. For non-normal data, nonparametric methods such as the Wilcoxon signed-rank test for paired samples and the Mann-Whitney U test for independent samples were used. The Kruskal-Wallis test was applied for data involving more than two groups. This approach ensures the use of suitable statistical tests on the basis of data distribution.

Hierarchical models: For hierarchical data, the lme4 package [44] was employed to generate linear mixed models, accounting for both fixed and random effects. This approach enabled us to account for the hierarchical structure of the data and mitigate the influence of potential confounding variables, thereby yielding results that are more precise and generalizable.

Presentation of results: The data are summarized as the means and standard deviations for continuous variables and as percentages for categorical variables. Statistical significance was set at *p* < 0.05, with all analyses conducted without missing data.

## 3. Results

### 3.1. Demographic Characteristics and Lifestyle

This study examined the body composition of a cohort of 432 Slovenian adults, of whom 287 were females (66%) and 145 were males (34%). Measurements were administered twice over the course of two years, culminating in a FU investigation comprising 389 adult participants, including 43 older adults. Within the female subgroup, 255 were categorized as adult females, and 32 were categorized as older adult females. In the male subgroup, 133 were designated young or middle-aged adult males, and 11 were categorized as older adult males. The geographical distribution of the participants was as follows: 26% from the capital city Ljubljana (including its environs), 25% from Upper Carniola, 17% from Styria, 16% from Lower Carniola, 9% from Inner Carinthia, and 3% each from Prekmurje and the Littoral Region (Figure 1). Furthermore, the education levels of the participants were categorized as follows: four years of secondary school (39%), higher education (24%), vocational education (16%), higher education professional (Bologna level 1) (12%), university programs (Bologna level 1) (7%), professional master’s degrees (Bologna level 2), doctorate degrees (Bologna level 3) (1%), and primary school (1%). With respect to smoking status, 16% were current smokers, 53% were ex-smokers, and 31% were nonsmokers. More than half (51%) of the participants engaged in physical activity at least three times per week, with a significantly greater proportion of males than females (*p* < 0.001).

### 3.2. Body Mass Index and Body Composition Status by Sex and Age

The mean BH values of the females and males were 165.9 cm ± 6.5 cm and 179.2 cm ± 7.5 cm (*p* < 0.001), respectively. Both adult females and males were taller than older adults were (166.3 cm ± 6.4 cm vs. 162.6 cm ± 7.0 cm and 179.4 cm ± 7.7 cm vs. 177.2 cm ± 4.8 cm, respectively; *p* < 0.001). Adult females had higher BM, BMI, FAT%, and FATM values and lower FFM% due to increased FATM. In addition, adult males had higher BM and BMI values, primarily due to higher FFM rather than FATM. The details are presented in Table 2.

### 3.3. Body Mass Index and FAT% Status According to BMI and FAT% Obesity Classification

Initially, 40.7% of individuals were classified as having overweight or obesity by BMI, increasing to 45.2% at the FU (*p* < 0.001). Notably, the percentage was significantly greater in males than in females (58.9% vs. 31.7% and 61.6% vs. 36.9%, respectively; both *p* < 0.001). The BMI classification revealed significantly fewer participants with obesity than the FAT% did (*p* = 0.013). Body mass index classification initially identified 13.4% of the participants as having obesity, increasing to 16% at the FU (*p* = 0.333), whereas FAT% initially identified 20%, increasing to 21.6% at the FU (*p* = 0.612). The BMI and FAT% status data for both classifications (initial vs. FU) for the entire sample (adults and older adults), including 95% confidence intervals, are presented in Appendix A.

Among adults, 38.2% were in the overweight or obese classifications by BMI initially, increasing to 42.4% at FU (*p* < 0.001), with higher rates in males than in females (55.7% vs. 29.2% and 59.0% vs. 33.9%, respectively; *p* < 0.001). Initially, 11.6% were classified as having obesity according to BMI, increasing to 14.2% at the FU (*p* = 0.007); FAT% classified 17.6% as having obesity initially, increasing to 19.5% at the FU (*p* < 0.810). The initial vs. FU BMI and FAT% data are shown in Table 3.

In the older adult cohort, 62.2% were initially in the overweight or obese classifications according to BMI, increasing to 68.4% at FU (*p* < 0.001), and this was higher in males (91.7%) than in females (51.5% or 60.5%). At FU, the rate of obesity according to BMI was 36.4% in females and 16.7% in males, although nonsignificantly different, and that of obesity according to FAT% was 39.4% in females and 41.7% in males, also nonsignificantly different.

Notably, 23% of females, 38% of males, and 28% of the adult cohorts were in the overweight category, whereas 24% of females, 75% of males, and 40% of older adults were in the overweight category.

### 3.4. Body Mass Index and FAT% According to Age

The age structure report illustrates a progressive distribution of BMI and FAT%, manifesting a notable increase in each age group. Specifically, young adults presented a lower BMI and FAT%. However, within each category, males had a higher BMI but a lower FAT% (Table 4). The median BMI and FAT% according to age, sex, and time of measurement (initial and FU assessments) are presented in Appendix A.

### 3.5. Trends and Variations Within Groups for Adults

Using linear mixed models, the study included 389 adults with 778 observations. Ageing led to a 0.21-unit annual weight gain; males weighed 18.83 units more than females did. The time of measurement increased the BM by 1.14 units. The random effects model showed a residual variance of 6.28 and an intercept variance of 188.70, with a 0.97 intraclass correlation. Fixed factors (age, sex, and time of measurement) explained 30.4% of the weight variability, whereas combined fixed and random effects explained 97.8%, highlighting individual differences. Each year of age increased BMI by 0.09 units, with the BMI of males being 2.46 units higher than that of females. The time of measurement increased BMI by 0.36 units, with 97% variance due to individual differences. Fixed factors accounted for 10.8% of BMI variability, with fixed and random effects accounting for 97.0%. Age increased the FAT% by 0.20% annually; males had a 7.41% lower FAT% than females did. The time of measurement increased the FAT% by 0.38%. Interindividual differences accounted for 93% of the FAT% variability, and fixed factors accounted for 27.7%, with both effects explaining 95%. Age did not significantly affect FFM; males had 19.88 units more FFM than females did, and each year of age increased FFM by 0.60. Interindividual differences explained 95% of the FFM variability, and fixed factors explained 65.5%, with both effects combined accounting for 98.1%. Each year of age, PhA decreased by 0.01 degrees; the PhA of males was 0.88 degrees higher than that of females. The time of measurement did not significantly affect PhA. Individual differences accounted for 83% of the PhA variance, and fixed factors accounted for 31.8%, with both accounting for 89.6%. The trends for the adults and the older adults are shown in Table 5 and Appendix A, respectively. The BMI, FAT%, FFM, and PhA distributions are shown in Appendix A.

## 4. Discussion

### 4.1. Main Findings

In this cohort study, trends in BMI and body composition among adult Slovenes were examined over a two-year period. The initial findings revealed that 40.7% of the participants were classified as having either overweight or obesity according to their BMI, and this percentage increased to 45.2% at the FU assessment. A comparable pattern was observed exclusively among adults. Importantly, this percentage was markedly greater in males than in females. The BMI classification estimated fewer participants with obesity than did the FAT%; however, the proportion of participants with obesity in both classifications remained similar between the initial and FU measurements. The hypothesis suggested an overall increase in overweight and obesity rates on the basis of both BMI and FAT% classifications, with the most significant increase observed among adult females. In contrast, adult males presented an increase in BM at the expense of FFM rather than FATM. Older adult females and males did not experience any major adverse changes. The linear mixed models highlighted the influence of sex, age, and time of measurement, emphasizing sex differences, the effects of ageing, and the importance of personalized health assessments.

### 4.2. Obesity Rates and BMI

Global obesity has overtaken underweight in the past four decades, shifting the overall health status of the human population [46]. The incidence of obesity in most European countries is increasing, being near 20% or higher, and healthcare costs and economic burdens are increasing [47,48]. Slovenia has the highest obesity rate among the 20 European countries studied recently, according to BMI classification (20.8%) [49].

Initially, 40.7% of the participants (adults and older adults) were in the overweight or obesity classifications on the basis of BMI; the prevalence rate significantly increased to 45.2% at the FU, with 61.6% for males and 36.9% for females. At the initial BMI measurements, 13.4% of the participants were in the obesity ranges; this prevalence rate increased nonsignificantly to 16% at the FU. Initially, 20.0% of the participants were categorized as in the obesity ranges according to the FAT%, increasing to 21.6% at the FU. Among adults, 38.2% were within overweight or obese categories according to BMI, increasing to 42.4% at FU, with 59.0% for males and 33.9% for females. Initially, 11.6% of the participants had obesity according to BMI, which increased to 14.2% at the FU. FAT% initially classified 17.6% as having obesity, increasing to 19.5% at the FU.

The results revealed a significant decline in adult females within two years of the initial measurement. Notably, 54% were aged 40 to 64 years. This review highlights how menopause affects body composition and fat distribution. Research shows that the menopausal transition impacts BMI and fat distribution, particularly during the perimenopausal and postmenopausal stages [50,51]. It is possible that increased health metrics result in part from midlife hormonal changes, such as fluctuating hormones and lower oestrogen levels in females. However, body composition data alone cannot provide definitive conclusions.

A previous study involving 151 individuals who switched to a plant-based diet yielded initial results similar to those of the current study. Specifically, 50% of the participants had overweight or obesity before adopting a plant-based diet [52]. Like the FU cohort study, the plant-based study sample included individuals from all Slovenian regions, with more females than males and more adults than older adults. However, these estimates regarding overweight or obesity rates slightly differ from the data gathered by the Slovenian national dietary survey (SI.Menu 2017/2018) on 780 adults and older adults, where 67% of the respondents were within the overweight or obesity ranges, with 59% being adults (68% males and 50% females) and 74% being older adults (79% males and 73% females) [17,53]. In Serbia, 2013 data showed that 60.5% of adults had overweight or obesity [54], aligning with the SI.Menu survey’s results [17,53]. A 2010 pan-European survey across 16 countries, including Croatia, Italy, Austria, and Hungary, revealed that 47.6% of European adults had overweight or obesity, with rates of 58.2% in Croatia, 38.5% in Italy, 44.5% in Austria, and 65.4% in Hungary [54]. These data, which are nearly 15 years old, are based on self-reported height and weight. Both the SI.Menu and pan-European surveys use BMI classification instead of direct FAT% measurements to calculate obesity rates.

### 4.3. Body Composition Assessment

Nutritional epidemiology faces challenges in the role of BMI in initial assessments, such as eligibility for various treatment modalities, e.g., bariatric surgery or semaglutide. Further evidence-based consensus is needed for guidelines and thresholds for different populations [27]. As the understanding of health and body composition evolves, the limitations of BMI for assessing health risks become more evident [55]. The calculation of BMI involves a quick assessment of height and weight but does not distinguish between FATM and FFM/LBM, risking health misclassification, particularly for individuals with high muscle mass or low muscle density [27,56]. The validity of BMI data is challenging, particularly in large samples, when height and weight are self-reported [54,57]. Relying solely on BMI and weight loss may offer limited health benefits and promote weight-related discrimination [28]. A U.S. study suggested that BMI may inaccurately reflect cardiometabolic disease, misclassifying approximately 75 million adults [29].

Dual-energy X-ray absorptiometry (DXA) is the gold standard for determining body composition because of its accuracy in differentiating bone mineral content, LBM, and FATM, outperforming BIA. However, the high cost, limited portability, and use of radiation of DXA limit its use. BIA is more practical, portable, cost-effective, and radiation-free, with good reproducibility and a reasonable correlation with DXA for overall FAT and FFM/LBM [58,59]. BIA accuracy can be affected by hydration and disease, leading to potential misestimation in different BMI groups [60,61,62,63]. A study of 35,000 UK Biobank participants revealed a strong correlation between BIA and DXA. BIA slightly underestimates the FATM and overestimates the FFM, with individual disparities linked to anthropometric measures [60]. Advancements in imaging have enhanced body fat measurements, including visceral adipose tissue (VAT) and ectopic fat in the heart and liver. DXA is practical, but MRI is the gold standard for precise VAT assessment, although it is more expensive and time-consuming [64,65]. Compared with BMI, advanced technologies such as BIA and DXA provide better insights into FAT distribution and composition. 

### 4.4. Age and ‘Normal BM’

In this analysis, adults were compared, and older adults were highlighted despite the limited sample size. Overweight was more common in older adults than in non-older adults (40% vs. 28%). This was observed mostly among males, with 75% of older adult males and 38% of adult males having overweight. This is due to the age distribution; most females aged 40–64 years were below the overweight threshold (24.9 kg/m^2^), whereas most males of the same age were within the overweight range (27.4 kg/m^2^).

Research indicates that a slightly elevated BMI may provide protective benefits for older adults. The “obesity paradox” suggests that individuals with an overweight BMI might have lower mortality risks and better functional and cognitive health than individuals with a lower BMI, especially those with an underweight BMI [66]. Older adults with a higher BMI often show better cognitive ability, physical independence, fewer depressive symptoms, and improved survival rates, although this does not always apply to individuals with obesity. Age-related factors such as obesity degree, muscle mass loss, fat distribution changes, and comorbidities can influence this pattern, making a slightly higher BMI potentially more beneficial than a normal BMI [66,67,68]. Lifestyle choices and pre-existing health conditions significantly impact outcomes in older adults, emphasizing the need for age-specific BMI criteria. Therefore, analyses of BMI in older adults should focus mainly on trends rather than health risks or benefits. Current BMI thresholds may not accurately reflect the health risks of excess fat in older adults, necessitating revised cut-off values. The inclusion of body composition metrics could improve health risk assessments for obesity in older age groups [66,69,70].

### 4.5. Linear Mixed Models

The general conclusions from the linear mixed models are as follows: (i) males tended to have higher BMI, FFM, and PhA but lower FAT%; (ii) in general, BMI and FAT% were higher in older adults, and PhA was lower, with no significant age effect on BM or FFM; and (iii) over a two-year period, BM, FFM, and FAT% decreased, but PhA remained unchanged. Overall, weight, BMI, FAT%, and FFM slightly increased with age and time of measurement, and PhA slightly decreased with age. The high intraclass correlation coefficient (ICC) values suggest that variability is due mainly to interindividual differences, indicating consistency within individuals.

The models emphasize the need for sex-specific health and fitness assessments, the varying impacts of ageing on body composition, and the necessity for regular monitoring. Individual variability underscores the importance of personalized health programs. In the cohort study, obesity was more prevalent in males than in females, which contrasts with global trends [2,71] and that reported in the SI.Menu study [53]. Interestingly, in the FU cohort and SI.Menu study, a higher proportion of adult males were classified as having overweight. Additionally, in the SI.Menu study, older adults typically had higher BMIs [53], which aligns with the findings of the present study. BMI increases with age due to changes in body composition, necessitating various health approaches [72,73].

### 4.6. Physical Activity and the BM

The initial study began shortly after the COVID-19 pandemic, during which lockdowns were implemented. The responses varied, with Slovenia expanding from regional to municipal partial lockdowns and advising low to moderate physical activity without specifying types. While the gyms were closed for several months, people’s activity levels fluctuated due to anxiety and uncertainty. Two years later, FU measurements revealed that BMI and body composition declined, raising questions about current activity levels as people returned to their routines. Half of the participants reported exercising at least three times weekly in the previous two months, but self-reported data are limited with regard to the definitive conclusions that can be drawn about overall activity trends.

A spring 2021 online survey in Slovenia with 1161 participants revealed that 80% of the participants maintained or increased their physical activity, mainly in natural settings [74]. It is likely that the physical activity was mainly low to moderate intensity instead of vigorous exercise, which is crucial for stimulating skeletal muscle. Additionally, in a recent review of 40 studies from 22 countries, excluding Slovenia but including Italy, physical inactivity, sedentary lifestyle, and poor eating habits were identified as common risk factors for obesity during the COVID-19 pandemic [75].

The degree of physical inactivity is increasing globally [76]. Managing obesity and improving body composition require a holistic approach that combines dietary changes with resistance and endurance exercises [77,78,79,80,81]. Choosing healthier foods can increase physical activity, and regular exercise can lead to better eating habits [82,83]. The synergy between physical activity and nutrition enhances a healthy lifestyle.

### 4.7. Possible Public Health Strategies

The FU study of Slovene adults revealed an imbalanced diet with low intake of complex carbohydrates and high consumption of ultra-processed foods. The intake of essential macronutrients such as carbohydrates and dietary fibre is insufficient, whereas the intake of total fat and saturated fatty acid is excessive. Micronutrient deficiencies were found for vitamins C, D, and E and calcium, and the sodium levels were above the recommended guidelines [11]. A national SI.Menu highlighted poor dietary quality with deficiencies in vitamin D, dietary fibre, and folate. Notably, 32% of adults and 46% of older adults, especially females aged 18 to 50 years, lack sufficient iron and vitamin B12 intake [15,16,17,84].

In terms of consumption according to food group, assessments indicate that the Slovenian diet is unbalanced, with excessive intake of meat, sugar, fat, and salt and insufficient intake of fruits, vegetables, and dairy products [85]. Research indicates that Slovenian meat consumption is approximately four times greater than the recommended level and six times greater than the Planetary Health Diet standard [85,86,87]. While some young adults are interested in reducing meat consumption, many of them continue to follow meat-heavy diets due to societal norms. This high meat intake, along with an overall unbalanced diet, poses serious health risks. Additionally, many young adults are unaware of the health and environmental impacts of excessive meat consumption, reducing their motivation to change their diets [88]. Research shows that individuals often lack sustainable dietary habits and are concerned with health, environmental issues, animal welfare, and dietary diversity [89]. Older adults are more proactive than younger individuals in reducing meat consumption [90]. Comprehensive strategies with cross-departmental collaboration and public engagement are urgently needed to address these dietary challenges.

In the FU study, 51% of the participants exercised at least three times weekly. In 2016, 56% of Slovenians met the WHO’s physical activity guidelines, with higher rates in males (59%) than in females (52%). Higher education correlated slightly with increased physical activity [91]. In terms of smoking status, 16% were current smokers, 53% were former smokers, and 31% were nonsmokers. In the national SI.Menu study, 55% of adults were nonsmokers [92]. These insights, along with poor dietary habits, highlight the urgent need for better lifestyle choices.

The widespread availability and marketing of a wide variety of inexpensive, convenient, energy-dense ultra-processed foods whose portion sizes are large and are high in fat, salt, and sugar and low in protein and fibre are major dietary drivers of increased obesity and common chronic diseases [93,94,95]. Additionally, low or irregular physical activity exacerbates hedonic behaviours and promotes overconsumption [93,96], and a firm government policy is essential to establish a healthy, profitable, fair, and sustainable food system that benefits everyone [19,20,97]. Nearly 1.8 billion adults worldwide are at risk of various diseases due to insufficient physical activity [76]. Engaging in regular physical activity and maintaining a healthy diet are typical methods to reduce excess BM/FATM and maintain FFM during BM loss [79,98,99,100], but this was not observed in the cohort of the present study.

The Slovenian Strategic Council for Nutrition has set 2023/24 activities to improve health and promote sustainable food. The government is encouraging healthier diets to benefit public health and the environment. A 2023–2025 action plan is being created to support national nutrition and physical activity programs, which are aligned with the Council’s initiatives and open for public review. The Council endorses the “Planetary Health Diet”, which consists mainly of plant-based products but includes some animal products and is linked to lower risks of diabetes, cardiovascular disease, cancer, and mortality, as shown by a recent systematic review [19,87,101].

### 4.8. Strengths and Limitations

This FU cohort study was the first to investigate body composition trends in the same group of Slovenian adults (two years after the COVID-19 pandemic). It extends the initial research, which was the only study on this topic at the population level in Slovenia. The measurements were consistent, taken within a month and at the same time of year as the initial study, thus minimizing seasonal influence. The initial sample was randomly selected, ensuring regional representation. The limitations of this study were the reduced FU adult sample (*n* = 432; 51%) compared with the initial large sample (*n* = 844) of the Slovene population (2,100,000 citizens in total). This sample included fewer older adults; therefore, despite random assignment and FU completion, the findings should be generalized with caution. BIA was used in this study; it is a cost-effective, radiation-free and portable method that is ideal for large ethnic groups [58,59] and is widely used in Slovenia at universities, medical institutions, rehabilitation centres, sports clubs and fitness centres [102] to assess body composition. In addition, compared with DXA, the 8-electrode BIA device offers accurate FAT% measurements [60,103,104], even when habitual physical activity is considered [105,106]. However, in the present study, a validity study of the specific BIA model versus DXA in various BMI categories is lacking [107,108]. While initial BIA measurements were used, gathering individuals for the same DXA device measurements was impractical and incomparable. Dietary intake was assessed in the initial study [11] but not during the FU, representing a limitation for interpretation and requiring reliance on speculation and literature. Although random sampling was used and all regions of Slovenia were covered, uneven representation limits the generalizability of the findings.

## 5. Conclusions

This is the first nationwide study on adult body composition in Slovenia. This representative cohort study revealed that from the initial study to the two-year FU, the proportion of adults with overweight and obesity increased from 40.7% to 45.2%, notably among females. Other groups, such as older females and males, presented no significant changes. Interestingly, the BM among adult males increased primarily because of FFM, not FAT. The study also revealed that BMI classification resulted in fewer participants with obesity than FAT% classification did, indicating the need for body composition analysis to determine obesity. The study highlighted the influence of sex, age, and time of measurement when assessing body composition, emphasizing the importance of personalized health assessments. This study highlights the need for joint efforts by governments, social organizations, and the food industry to encourage lifestyle changes through education and a supportive environment for healthy living.

## Figures and Tables

**Figure 1 nutrients-16-04123-f001:**
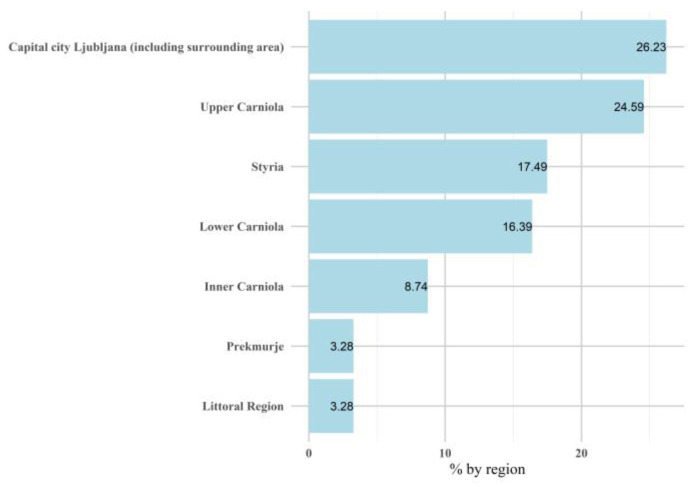
Regional distribution of participants.

**Table 1 nutrients-16-04123-t001:** Obesity classification according to BMI and FAT% [32,33].

BMI Categories		FAT% Categories	
Status	BMI (kg/m^2^)	Status	FAT%
Underweight	<18.5	Female—normal	<35
Normal body mass	18.5–24.9	Female—obesity	>35
Pre-obesity/overweight	25.0–29.9	Male—normal	<25
Obesity class I	30.0–34.9	Male—obesity	>25
Obesity class II	35.0–39.9		
Obesity class III	Above ≥ 40		

BMI = body mass index and FAT% = total body fat percentage.

**Table 2 nutrients-16-04123-t002:** Body mass index and body composition status by sex and age.

Age (Years)	All 18–64			All 65+			Females 18–64			Females 65+			Males 18–64			Males 65+		
	IN	FU	*p* Value	IN	FU	*p* Value	IN	FU	*p* Value	IN	FU	*p* Value	IN	FU	*p* Value	IN	FU	*p* Value
BM (kg)	72.6 ± 16.7	73.7 ± 16.7	**<0.001**	75.4 ± 13.0	76.3 ± 12.9	0.170	66.2 ± 12.4	67.4 ± 12.5	**<0.001**	71.7 ± 12.2	72.4 ± 11.8	0.464	85.1 ± 17.0	86.1 ± 17.0	**0.032**	85.9 ± 9.3	87.1 ± 9.3	0.999
BMI (kg/m^2^)	24.8 ± 4.7	25.1 ± 4.6	**<0.001**	27.2 ± 3.9	27.5 ± 3.8	0.120	23.9 ± 4.4	24.3 ± 4.3	**<0.001**	27.1 ± 4.4	27.4 ± 4.3	0.261	26.4 ± 4.8	26.7 ± 4.9	**0.039**	27.3 ± 2.0	27.6 ± 2.1	0.999
FATM (kg)	18.7 ± 9.2	18.8 ± 9.1	**<0.001**	23.5 ± 7.8	23.9 ± 7.6	0.999	18.6 ± 8.5	19.2 ± 8.4	**<0.001**	24.8 ± 8.0	25.2 ± 7.8	0.999	17.6 ± 10.3	17.9 ± 10.2	0.999	19.7 ± 5.7	20.3 ± 6.0	0.999
FAT%	24.5 ± 8.1	24.9 ± 8.0	0.053	30.9 ± 7.5	31.1 ± 7.3	0.999	27.0 ± 7.5	27.5 ± 7.4	**0.048**	33.9 ± 5.7	34.0 ± 5.5	0.999	19.7 ± 6.8	19.8 ± 6.8	0.999	22.6 ± 5.3	23.0 ± 5.2	0.999
FATM/BH (kg/cm)	10.7 ± 5.4	11.0 ± 5.3	**<0.001**	14.1 ± 4.8	14.4 ± 4.7	0.885	11.2 ± 5.1	11.6 ± 5.0	**<0.001**	15.3 ± 4.8	15.5 ± 4.7	0.999	9.8 ± 5.7	10.0 ± 5.7	0.999	10.9 ± 3.0	11.4 ± 3.2	0.999
FFM (kg)	54.3 ± 11.6	54.9 ± 11.7	**<0.001**	51.9 ± 10.1	52.4 ± 10.1	0.972	47.6 ± 5.5	48.2 ± 5.6	**<0.001**	46.9 ± 5.3	47.3 ± 5.1	0.999	67.5 ± 8.7	68.1 ± 8.8	**0.012**	66.3 ± 5.8	66.8 ± 5.4	0.863
FFM (%)	75.5 ± 8.1	75.1 ± 8.0	0.059	69.1 ± 7.5	68.8 ± 7.3	0.999	73.0 ± 7.5	72.5 ± 7.4	**0.046**	66.1 ± 5.7	65.9 ± 5.5	0.999	80.3 ± 6.8	80.2 ± 6.8	0.999	77.4 ± 5.3	77.0 ± 5.2	0.999
FFM/BH (kg/cm)	31.6 ± 5.4	32.0 ± 5.4	**<0.001**	31.0 ± 4.7	31.3 ± 4.7	0.999	28.6 ± 2.9	28.9 ± 2.9	**<0.001**	28.8 ± 2.9	29.0 ± 2.7	0.999	37.5 ± 4.0	37.9 ± 40	**0.025**	37.4 ± 2.6	37.6 ± 2.5	0.999
TBW (kg)	38.8 ± 8.3	39.2 ± 8.4	**<0.001**	36.3 ± 7.0	36.7 ± 6.9	0.999	34.0 ± 4.0	34.4 ± 4.0	**0.002**	33.0 ± 3.8	33.3 ± 3.6	0.935	48.0 ± 6.4	48.6 ± 6.7	**0.018**	45.9 ± 4.6	46.3 ± 4.3	0.999
PhA (°)	5.9 ± 0.7	5.9 ± 0.7	0.999	5.2 ± 0.6	5.2 ± 0.6	0.999	5.6 ± 0.5	5.6 ± 0.6	0.999	5.2 ± 0.5	5.2 ± 0.6	0.999	6.5 ± 0.6	6.5 ± 0.6	0.999	5.3 ± 0.7	5.3 ± 0.7	0.863

The data are presented as the means ± standard deviations. Statistically significant values are shown in bold. IN = initial, FU = follow-up, BH = body height, BM = body mass, BMI = body mass index, FAT% = total body fat percentage, FATM = total body fat mass, FFM = fat-free mass, TBW = total body water, and PhA = whole-body phase angle.

**Table 3 nutrients-16-04123-t003:** Body mass index and FAT% status of adults according to BMI and FAT% obesity classification.

		Initial (*n* = 389)			FU (*n* = 389)		
Parameter		All	Females	Males	All	Females	Males
According to BMI classification (%) ^†^							
Normal (BMI 18.5–24.9 kg/m^2^)		61.8	70.9	44.2	57.6	66.1	41.1
Overweight (BMI 25–29.9 kg/m^2^)		26.6	19.9	39.5	28.2	23.1	38.0
Obesity 1 class (BMI 30–34.9 kg/m^2^)		8.7	7.2	11.6	11.1	8.0	17.1
Obesity 2 class (BMI 35–39.9 kg/m^2^)		1.6	1.2	2.3	1.8	2.0	1.6
Obesity 3 class (BMI >40 kg/m^2^)		1.3	0.8	2.3	1.3	0.8	2.3
*p* value		**<0.001**			**<0.001**		
According to BMI obesity classification (%) ^†^							
Normal		88.4	90.8	83.7	85.8	89.2	79.1
With obesity		11.6	9.2	16.3	14.2	10.8	20.9
*p* value		**0.040**			**0.009**		
According to FAT% obesity classification (%) ^†^							
Females	<35% (normal)		84.1			80.9 19.1	
	>35% (obesity)		15.9				
Males	<25% (normal)			79.1			79.8
	>25% (obesity)			20.9			20.2
All	Normal	82.4			80.5		
	Obesity	17.6			19.5		
*p* value		0.226				0.891	

Statistically significant values are shown in bold. ^†^ Body mass index (BMI) and FAT% obesity classifications by the WHO [33,45]. BMI = body mass index and FAT% = total body fat percentage.

**Table 4 nutrients-16-04123-t004:** Body mass index and FAT% according to age structure.

	Initial				Follow-Up			
*Females*								
Age category (years)	18–39 (*n* = 99)	40–64 (*n* = 156)	≥65 (*n* = 32)	*p* value	18–39 (*n* = 99)	40–64 (*n* = 156)	≥65 (*n* = 32)	*p* value
BMI (kg/m^2^)	23.0 ± 4.0	24.5 ± 4.5	26.9 ± 4.4	<0.001	23.4 ± 4.1	24.9 ± 4.4	27.2 ± 4.2	<0.001
FAT%	24.4 ± 7.5	28.5 ± 7.1	33.9 ± 5.5		24.6 ± 7.5	29.2 ± 6.8	33.9 ± 5.4	
*Males*								
Age category (years)	18–39 (*n* = 56)	40–64 (*n* = 78)	≥65 (*n* = 11)	*p* value	18–39 (*n* = 56)	40–64 (*n* = 78)	≥65 (*n* = 11)	*p* value
BMI (kg/m^2^)	25.3 ± 4.7	27.1 ± 4.8	27.6 ± 2.2	<0.001	25.6 ± 4.7	27.4 ± 4.8	28.0 ± 2.3	0.002
FAT%	17.5 ± 6.7	21.0 ± 6.5	23.2 ± 5.5		17.9 ± 6.7	21.0 ± 6.5	23.7 ± 5.5	

BMI = body mass index and FAT% = total body fat percentage.

**Table 5 nutrients-16-04123-t005:** Trends and variations within groups for adults.

	BM		BMI		FAT%		FFM		PhA	
Predictors	Estimates	*p* Value	Estimates	*p* Value	Estimates	*p* Value	Estimates	*p* Value	Estimates	*p* Value
Intercept	55.83	**<0.001**	19.81	**<0.001**	18.07	**<0.001**	46.30	**<0.001**	6.18	**<0.001**
Age	0.21	**<0.001**	0.09	**<0.001**	0.20	**<0.001**	0.02	0.593	−0.01	**<0.001**
Sex (male)	18.83	**<0.001**	2.46	**<0.001**	−7.41	**<0.001**	19.88	**<0.001**	0.88	**<0.001**
Time of measurement	1.14	**<0.001**	0.36	**<0.001**	0.38	**0.004**	0.60	**<0.001**	0.01	0.450

Statistically significant values are indicated in bold. BM = body mass, BMI = body mass index, FAT% = total body fat percentage, FFM = fat-free mass, and PhA = whole-body phase angle.

## Data Availability

The data used to support the findings of this study are included within the article.

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
