# Peer review of "Body Composition Trend in Slovene Adults: A Two-Year Follow-Up"

_nutrients, 2024, doi:10.3390/nu16234123_

Round 1
Reviewer 1 Report
Comments and Suggestions for Authors
authors)
STRUCTURE
- The manuscript is correctly structured. But there are some points to consider:
o Tables: including the meanings of some abbreviations, such as FU and BL.
TITLE AND ABSTRACT
- The title mentions the type of research that has been carried out.
- The summary is well structured, listing all parts.
- Here are some recommendations for providing a summary in a clear, concise and scientifically rigorous manner:
o Comparison between BMI and FAT%:
For clarity, detail the purpose and findings of the comparison between BMI and FAT%. Currently, the sentence could be confusing. For example:
§ ‘Comparing BMI and FAT% classifications, the BMI approach categorised fewer participants as obese in both assessments, although the proportion obese remained stable across the two years regardless of the method used.’
o Clarification of the main objective:
The first sentence should clearly state the purpose of the study. Instead of ‘This follow-up study of 432 adults from Slovenia aimed to reevaluate the prevalence of obesity based on the World Health Organization's body mass index (BMI) and total body fat percentage (FAT%) classifications two years after an initial population-based cross-sectional study,’ you could simplify it to something like:
§ ‘This study reassesses the prevalence of obesity among 432 Slovenian adults, two years after an initial population-based cross-sectional study, using the BMI and total body fat percentage (FAT%) classifications.’
INTRODUCTION
- Line 51: “Additionally, our initial study revealed that 42% of adults and 64% of elderly…”. Remember not to speak in the first-person plural.
- Some aspects of improvement are also proposed for this section:
o Revision of style and conciseness.
Recommendation: Restructure long sentences and eliminate repetitions. In addition, use consistent terms (e.g., always refer to ‘obesity and overweight’ rather than interchange terms) and reduce unnecessary or redundant quotations. Example of restructuring:
§ Instead of ‘According to a 2022 report by the World Health Organization (WHO), 59% of adults in Europe are overweight or obese, highlighting an urgent epidemic [10],’ it can be simplified to ‘Some 59% of adults in Europe are overweight or obese, according to WHO, reflecting an alarming trend [10].’
- As for the specific objectives or hypotheses, it appears that the objectives and hypotheses of the study are well defined.
MATERIAL AND METHODS
- Study design. The research presents the key elements of the study design at the beginning of this section.
- In the participants' section, researchers should describe the relevant setting, locations and dates, including recruitment, exposure, monitoring and data collection periods.
- Eligibility criteria are stated, but sources and methods of participant selection are not. Incorporate the latter.
- Line 150: ‘A sociodemographic questionnaire collected...’: What type of questionnaire is used? Has it been previously validated for this population? It would be interesting to provide some reference of this questionnaire in this type of population.
o Statistical analysis
- To improve clarity and rigour in this statistical analysis section, you can make the following adjustments and improvements.
1. Structure and conciseness
Recommendation: Organise the paragraph into sections so that each step of the analysis is easy to follow. Using subheadings or transition phrases such as ‘Data preparation’, ‘Comparison tests’, and ‘Hierarchical models’ helps to improve structure. Example:
§ Data preparation: ‘We used tidyverse [37] for data manipulation and visualisation, leveraging its cohesive framework for efficient workflow. janitor [38] facilitated data cleaning, generating tidy sets ready for analysis.’
§ Descriptive statistics: ‘Arsenal [39] was used to obtain detailed summaries and descriptive statistics for key study variables.’
§ Analysis of differences in obesity: ‘To compare obesity between methods (BMI vs. FAT%), we applied a two-sample test for equal proportions with continuity correction.’
2. Explanation of statistical tests and correction methods
Recommendation: Briefly explain the rationale for each statistical test and correction method used. This provides context as to why these tests were chosen and increases the transparency of the analysis. Example:
§ ‘To compare proportions of obesity between classification methods (BMI vs. FAT%), a two-sample test for equal proportions with continuity correction, suitable for large samples, was employed. In tests of multiple significance, p-values were adjusted with the Benjamini-Hochberg method to control for the false discovery rate’.
3. More detail on parametric and non-parametric tests
Recommendation: Clarify why and when non-parametric tests were used, specifying how it was determined whether the data met the assumptions of normality. Example:
§ ‘T-tests were applied in cases where the data showed a normal distribution; in contrast, non-parametric signed-rank tests for paired samples were used when the data did not meet the assumptions of normality, thus ensuring a robust assessment.’
3. Description of the presentation of results
Recommendation: Mention how statistical results are presented (means, standard deviation) and make it more explicit that a significance threshold was followed. Example:
§ ‘Results are presented as means and standard deviations for continuous variables and percentages for categorical variables, with a significance threshold of p < 0.05 being set.’
RESULTS
- Table 3. Enter the meaning of the abbreviation FU.
- To improve the results section and to achieve greater clarity, accuracy and usefulness of the research findings, the following strategies can be applied and detailed with examples:
1. Breakdown of variables and descriptives:
Suggested improvement: The descriptive analysis could benefit from including segmented tables and displays that show mean data and standard deviations not only by gender, but also by age range and region, facilitating a detailed reading of differences between subgroups. Example:
§ Present a table detailing demographic characteristics (such as height, weight, BMI) for each age group and region, as well as bar charts or box plots to visualise regional and age trends. This would help to identify patterns of body composition based on geographical context.
- Use of Advanced Statistical Models to Control Covariates
- Proposed improvement: Include covariates such as education level and smoking in the mixed models. This could help to understand whether certain demographic or lifestyle characteristics influence BMI and body fat percentage (%BFAT).
- Example: Conducting linear mixed models that include education level and smoking as covariates. These models could reveal, for example, whether higher educational levels correlate with lower %FAT or whether smoking impacts FFM (fat-free mass).
- 6. Obesity Classification Analysis with Confidence Intervals
- Proposed improvement: Including confidence intervals when reporting obesity classifications (BMI and %FAT) would increase the robustness of the analysis and provide a measure of the precision of the results.
- Example: Calculate the 95% confidence interval for each obesity and overweight category at baseline and follow-up. This can help to assess the precision of the observed changes in obesity rates.
- 8. Segmentation by Educational Level and Smoking Status in Comparisons of BMI and %FAT.
- Proposal for Improvement: Conduct comparative analyses that show differences in BMI and %FAT between different educational levels and smoking status.
- Example: Assess whether %FAT is higher in individuals with secondary education versus those with university education or whether non-smokers have a significantly lower %FAT than ex-smokers and current smokers.
- 10. More Detail in the Presentation of Comparison Tables and Graphs
- Suggested improvement: In tables, results could benefit from visual indicators, such as bold p-values or highlighting significant differences. Also, adding scatter plots with trend lines for BMI, %FAT and FFM would help to observe the temporal relationship.
- Example: A table with p-values and effect estimates in separate columns, using colours to highlight significant comparisons. For trend graphs, regression lines for each gender group would show changes over time in a visually accessible way.
o Including these additional elements would enrich the results and facilitate the practical interpretation of the findings, as well as better contextualise the changes in a public health framework.
DISCUSSION
- To enhance the discussion of the study, the following points could be added:
o International context and global comparisons:
Compare the findings of this study with data from other countries, especially within the European Union and other developed countries, to place the Slovenian results in a broader context and highlight how trends in overweight and obesity in Slovenia compare with those observed in other populations.
o Implications of post-pandemic lifestyle changes:
To delve into how changes in lifestyle and physical activity levels during and after the COVID-19 pandemic may have influenced the observed weight and body composition trends, particularly for young adults and women, who showed increased rates of obesity.
o Limitations of Exclusive Use of BIA: Although a useful and practical tool, the use of BIA to assess body composition has certain limitations, such as variability in accuracy with respect to visceral fat and fat distribution. It would be useful to discuss how the incorporation of other methods, such as DXA or MRI, could improve the accuracy of measurements in future research.
- On the other hand, how does the impact of ultra-processed food affect the Slovenian population? Discuss how the consumption of ultra-processed foods, mentioned in the global context, might be affecting the Slovenian population specifically, and explore whether these patterns are contributing to the increase in obesity in the country. Mention health policies that could be implemented to reduce the consumption of ultra-processed foods in the population.
- It is also proposed to make an association between physical activity and body mass. This implies including an analysis of the relationship between the level of weekly physical activity and the changes in body composition observed in the study. Specifically, to analyse whether those who reported higher levels of physical activity showed differences in fat-free mass (FFM) or body fat (FAT) compared to more sedentary participants.
- What are the implications of the discrepancies between obesity classification based on BMI and that based on % body fat? Point out how this difference could affect the diagnosis and treatment of obesity and highlight the importance of implementing more comprehensive assessments of body composition in clinical practice.
- Include possible biases and effect of sample reduction on follow-up, discussing how the reduced follow-up sample (51%) might affect the results and generalisability of the findings. Also, reflect on how participants who remained in the study might differ in key characteristics (such as motivation to maintain a healthy lifestyle) compared to those who dropped out, which could influence the results.
- Given the increase in obesity rates in Slovenia, discuss possible public health strategies that could be implemented, such as nutrition education programmes, promotion of physical activity, and controlling the availability of ultra-processed foods, targeting specific groups such as women and older people.
- Adding these points to the discussion would provide a more comprehensive, better contextualised analysis with clearer implications for public health policy and clinical practice in Slovenia and other similar contexts.
CONCLUSION
- The conclusions section does appear, but it would be pertinent to rename some conclusions and limitations briefly and in line with aspects of improvement for future research.
Reviewer 2 Report
Comments and Suggestions for Authors
This paper emphasized the necessity of personalized health assessments and the importance of regular body composition measurements in evaluations of adult obesity more accurate. This is important from the view of clinical assesments as well as for policies for the public health.
The main problem is that the numbers of respondents in the groups being observed are not specified (eg how many people were in the 65+ group). if the number is negligible - it is more appropriate to remove them from this study, because a BMI of 18.5-25 kg/m2 is not appropriate for the elderly as "normal" (https://www.e-agmr.org/upload/pdf/ agmr-22-0012.pdf) as there are percentile curves for young people up to 20 years of age that express nutrition.
However, clarifications of some important issues are needed.
What is problematic is that the discussion did not even touch on the mentioned issues of age and "normal BMI", and accordingly the results of the model in table 3 are influenced (while age shows a significant influence - which is not objective).
The coverage of 66% of women is acceptable (although it is not representative of the ratio of the female population compared to the male population in Slovenia), however, if the majority of these women are in the premenopause/menopause transition, and this data is not taken into the model - it is expected that the "time of measurement" more significant impact on the female population, if the increase in BM and BMI is observed.
It is necessary to elaborate on ALL the listed doubts in the paper itself (as well as indicate the number of respondents in each individual group), therefore I suggest a major revision.
Sincerely
Reviewer 3 Report
Comments and Suggestions for Authors
The authors should be congratulated for doing a good job with this manuscript. This analysis was conducted carefully, with good discussion of decisions made for data trimming and selection of appropriate statistical methods.
The document is written clearly, and findings are reported in the tables in an understandable manner.
Implications for clinical practice are discussed rather well. Perhaps another paragaph could be added to accentuate recommendations for policymakers related to regulatory intervention or other policy measures.
As a minor point, in lines 154-155 ("For our statistical analysis, we used R 4.3.1 and RStudio 2024.04.1 to build 748 with various packages for effective data handling and analyses") the meaning of "748" needs to be explained. Is this related to ICD-9 code 748 for congenital anomalies of the respiratory system?
Round 2
Reviewer 2 Report
Comments and Suggestions for Authors
In the revised version, the authors objectively discussed the limitations, and in particular commented on the study, which differed methodologically from the others. Now the work is at the level of an objective scientific study and I agree with the publication.
Sincerely
Author Response
Dear,
We thank you for all the comments and suggestions and for the accepted corrections.
With respect,